# XRD and TG-DTA Study of New Phosphate-Based Geopolymers with Coal Ash or Metakaolin as Aluminosilicate Source and Mine Tailings Addition

**DOI:** 10.3390/ma15010202

**Published:** 2021-12-28

**Authors:** Dumitru Doru Burduhos Nergis, Petrica Vizureanu, Andrei Victor Sandu, Diana Petronela Burduhos Nergis, Costica Bejinariu

**Affiliations:** 1Faculty of Materials Science and Engineering, Gheorghe Asachi Technical University of Iasi, 700050 Iasi, Romania; doru.burduhos@tuiasi.ro (D.D.B.N.); sav@tuiasi.ro (A.V.S.); burduhosndiana@yahoo.com (D.P.B.N.); 2Materials Science and Engineering Section, Technical Sciences Academy of Romania, Dacia Blvd 26, 030167 Bucharest, Romania; 3Romanian Inventors Forum, St. P. Movila 3, 700089 Iasi, Romania

**Keywords:** phosphate-based geopolymers, thermal behaviour, thermogravimetry-differential thermal analysis, phase analysis

## Abstract

Coal ash-based geopolymers with mine tailings addition activated with phosphate acid were synthesized for the first time at room temperature. In addition, three types of aluminosilicate sources were used as single raw materials or in a 1/1 wt. ratio to obtain five types of geopolymers activated with H_3_PO_4_. The thermal behaviour of the obtained geopolymers was studied between room temperature and 600 °C by Thermogravimetry-Differential Thermal Analysis (TG-DTA) and the phase composition after 28 days of curing at room temperature was analysed by X-ray diffraction (XRD). During heating, the acid-activated geopolymers exhibited similar behaviour to alkali-activated geopolymers. All of the samples showed endothermic peaks up to 300 °C due to water evaporation, while the samples with mine tailings showed two significant exothermic peaks above 400 °C due to oxidation reactions. The phase analysis confirmed the dissolution of the aluminosilicate sources in the presence of H_3_PO_4_ by significant changes in the XRD patterns of the raw materials and by the broadening of the peaks because of typically amorphous silicophosphate (Si–P), aluminophosphate (Al–P) or silico-alumino-phosphate (Si–Al–P) formation. The phases resulted from geopolymerisation are berlinite (AlPO_4_), brushite (CaHPO_4_∙2H_2_O), anhydrite (CaSO_4_) or ettringite as AFt and AFm phases.

## 1. Introduction

Globally there is a continuing concern for the research and development of green materials for civil engineering, in particular for the replacement of those based on Ordinary Portland Cement (OPC) [1,2]. Thus, it is essential to improve both the conventional technologies for the exploitation of natural resources and the technologies for obtaining concrete based on OPC. The purpose of these changes is to convert existing cement plants into facilities suitable for the manufacture of green concrete, such as geopolymers [3]. Geopolymers are eco-friendly materials created through the geopolymerisation chemical reaction which occurs after mixing an aluminosilicate source with an aqueous solution. This multiple-stage reaction consists of (1) dissolution of the aluminosilicate source in acidic medium, (2) Si–O–Al network and gel formation and (3) formation of a geopolymer structure [4].

A comprehensive review of alkali-activated geopolymers was conducted by Almutairi et al. [5], according to their study, multiple aluminosilicate wastes, such as red mud, ground granulated blast slag, rice husk ash, fly ash and glass powder can be used as raw materials for the alkali-activated cement. Moreover, the resulting material will exhibit high chemical stability and an 80% reduction in CO_2_ emissions compared with their main competitor (OPC-based materials). In another study [6], another aluminosilicate, volcanic pumice dust, was mixed with OPC or cement kiln dust to obtain high-performance geopolymers. In terms of compressive strength, the optimum mixture was that of volcanic pumice dust and cement kiln dust, while the porosity and water absorption at a curing age of 28 days were almost the same for all mixtures. In another study, Arpajirakul S. et al. [7] evaluated the effect of urea-Ca^2+^ addition on soft soil stabilisation by microbially-induced calcite preparation technique. According to their study, the compressive strength can be increased up to 18.50% at an optimum urea addition rate of 7.5 mmol/h.

The interest in geopolymer development has been boosted by their versatility, association with a multitude of aluminosilicate sources, production parameters and potential activators [8]. However, the multitude of production process parameters as well as the variety of sources of raw materials, slows down the market transition and industrial development of these materials. Another barrier in their industrial development is the high price of activation solutions, especially sodium silicate [9]. Therefore, recent studies in the field have been focused on the discovery and development of new activation solutions which are cheap and have a low environmental impact. Thus, phosphoric acid has become the main competitor of alkaline activators used to date in geopolymers [10,11]. However, the replacement of alkaline solutions with phosphoric acid has led to a major change in the formation mechanism of geopolymers; therefore, the obtained materials may have chemical, structural, thermal and mechanical characteristics which are different from those of alkaline-activated geopolymers. Accordingly, to reach the industrial development potential of phosphoric acid-based geopolymers, it is necessary to study all of their characteristics.

Wang Y.S. et al. [12] successfully synthesized phosphate-based geopolymers using silica fume mixed with metakaolin as an aluminosilicate source and monoaluminium phosphate for the activator solution. According to their study, at an Al/P ratio of one, the optimum compressive strength was obtained due to the formation of SiO_2_·Al_2_O_3_·P_2_O_5_·nH_2_O and AlH_3_(PO_4_)_2_·3H_2_O. Moreover, these main reaction products of geopolymerisation will be converted to berlinite when exposed to high temperatures. In another study [13], the authors evaluated the thermal behaviour and water resistance of geopolymers obtained from a mixture of metakaolin with phosphoric acid (10 M) as an activator. According to their study, the geopolymers with a Si/P ratio of 0.82, cured at 60 °C for 24 hours and aged in air for 28 days will exhibit a 50% higher compressive strength than those aged in water. The phenomenon was correlated with the hydrolysis of Si–O–P bonds during ageing. However, even in these curing conditions, the obtained geopolymers showed comparable mechanical properties with Ordinary Portland Cement (OPC) materials; therefore, these materials can be used in civil engineering applications.

Bai C. et al. [14] synthesized foams with a homogeneous microstructure by mixing metakaolin with phosphoric acid (85 wt.%) and water at a molar ratio of H_3_PO_4_/Al_2_O_3_ = 1.8, SiO_2_/Al_2_O_3_ = 2.4 and H_2_O/H_3_PO_4_ = 6.7, respectively. The geopolymer foam produced had a total open porosity as high as 76.8 vol%, and compressive strength of 0.64 MPa. In addition, when exposed to high temperature a 6.4% shrinkage and close to 90% weight loss was observed. The shrinkage was associated with the mesopores decreasing and densification due to dehydration of the structure, yet they concluded that the obtained material is suitable to replace the conventional highly porous materials in industrial applications.

Zribi M. et al. [15] evaluated the structure of phosphate-based geopolymers by combining four different techniques (magic angle spinning nuclear magnetic resonance (MAS-NMR), Fourier transform infrared spectroscopy (FTIR), X-ray diffraction powder (XRD) and scanning electron microscopy (SEM)). According to their study, the material obtained by mixing metakaolin with phosphoric acid at an Al/P ratio of one and cured at 60 °C for 24 h exhibited an amorphous structure composed of an aluminium phosphate geopolymeric network dispersed in a base created from Si–O–Si, Si–O–Al and Si–O–P units.

According to Djobo J.N.Y. et al. [16], the phosphate geopolymers are obtained due to the reaction of Al^3+^ ions with the proton H^+^ and H_2_PO_4_—species resulting from the deprotonation of commercial H_3_PO_4_. Moreover, when the aluminosilicate source is rich in different types of metal compounds besides Al^3+^, the obtaining reaction involves Fe^2+^/Fe^3+^, Ca^2+^ and Mg^2+^ ions dissolution. The resulting ions will react with the phosphate species in the following order: Ca^2+^ = Mg^2+^ > Al^3+^, Fe^2+^/Fe^3+^, resulting in calcium phosphate, magnesium phosphate aluminophosphate, silico-aluminophosphate, silicophosphate and iron phosphate phases.

One of the reasons why the interest in the development of geopolymers activated with acid solutions based on potassium is increasing, is the superior compressive strength of the resulting materials, compared to those which are alkaline-activated [17]. Moreover, the economic aspects also tip the scales in favour of the acid-activated ones, this aspect is due to the fact that the sodium hydroxide solutions do not present a sufficiently high geopolymerization potential when used alone. To develop suitable mechanical properties for civil engineering applications, the alkaline activator most often consists of a mixture of sodium hydroxide solution and sodium silicate. The sodium silicate is in a larger amount than the sodium hydroxide solution. This aspect greatly influences the price of the final product due to the high purchase price of sodium silicate. Given that the industrial development of a product is limited by its price, obtaining a cheap alternative to geopolymers and their development is essential for the transition from conventional materials (OPC-based concrete) to sustainable materials (geopolymer concrete).

So far it has been observed that in the case of potassium-based geopolymers, the compressive strength developed over time is higher than that of sodium-based geopolymers.

The effect of the Si–Al ratio on the mechanical and structural characteristics of geopolymers has been reported in a multitude of studies [18,19]. According to these studies, it was observed that the best properties are obtained for Si/Al between 1.5 and 1.9. It has also been observed that another chemical ratio with a significant influence on geopolymers is that between Na/Al [20]. Its optimal value is close to one, however, a decrease in the ratio leads to the production of a structure with high porosity, and its increase results in an improvement of the compressive strength. In the case of acid-activated geopolymers, the chemical ratio of primary interest is that between P and Al. According to previous studies [12,21], it influences the mechanical properties of geopolymers in a similar way to the ratio of Na to Al, but to our knowledge, no study has evaluated the thermal behaviour on ambient cured geopolymers activated with phosphoric acid in different P/Al ratios.

Multiple previous publications have focused on the effects of phosphoric acid activation on the mechanical characteristics of geopolymers. Moreover, most of the studies use metakaolin as a raw material because it has a simple chemical composition compared with other precursors. Therefore, there is a lack of information on the thermal behaviour and phase transition of phosphate-based geopolymers, especially on those which use coal ash or other by-products as aluminosilicate sources. This study aims to evaluate the influence of curing parameters and phosphate acid concentration on the thermal behaviour and phase transition of coal ash-based geopolymers with mine tailings content. As presented in [22,23], the minerals containing sulfides, such as pyrite, pyrrhotite and arsenopyrite can be oxidized when mixed with water or oxygen. Therefore, harmful metals can be released into the environment. Another advantage of using mine tailings in geopolymer development is related to their capacity for immobilizing harmful species during the hardening process. During the formation of ettringite, different elements can be replaced with others that have a similar radius and oxidation state. Accordingly, multiple metals, the harmful one included, will be encapsulated into the structure of the geopolymer, as follows: Ca^2+^ will be replaced by Mg^2+^, Co^2+^ and Zn^2+^; Al^3+^ will be replaced by Cr^3+^, Sr^3+^ and Fe^3+^; and SO_4_^2−^ will be replaced by oxyanions of Cr and As.

Accordingly, this study investigates the effect of room temperature hardening on phase transition during geopolymerisation in five types of geopolymers obtained by mixing three types of raw materials (coal ash, metakaolin and mine tailings). Moreover, the thermal behaviour of these materials was analysed up to 600 °C.

## 2. Materials and Methods

The obtained geopolymer was manufactured by mixing the raw material with a commercially available acid solution of o-phosphate (H_3_PO_4_) with 85 wt.% solid content. The solid to liquid ratio was optimized to assure an Al/P ratio of 1, for both types of geopolymers.

### 2.1. Materials

#### 2.1.1. Coal Ash (CA)

In this study, local coal ash from CET II Holboca, Iasi, Romania was collected and processed. To ensure experimental repeatability, the collected powder was firstly dried in a chamber with a static atmosphere, i.e., without ventilation, to avoid fine particle removal, and was secondly sifted to remove the particles with a diameter above 100 µm. The drying method is common and is presented in the literature [20], while the sifting method has been presented in a previous study [4]. The coal ash used in the study had a particle size distribution of 3.4 μm (d50) and a specific surface area of 1.6 m^2^/g as determined by a Coulter LS 200 laser scattering particle size distribution analyser (Beckman Coulter Inc., Pasadena, CA, USA). In addition, the bulk density of coal ash was 2.16 ± 0.01 g/cm^3^ evaluated with a Le Chatelier densimeter (Recherches & Realisations Remy, Montauban, France).

According to XRF analysis, the coal ash used in this study belongs to class F fly ashes. From a microstructural point of view (Figure 1), the aluminosilicate waste collected is a mixture of fly ash and bottom particles, as both spherical and irregular porous particles can be seen.

Coal ash is a mineral residue resulting from coal combustion in thermal power plants, it has small particles in the range of 0.01 to 300 µm. The chemical composition of coal is critical since it influences the ultimate properties of geopolymers. Silicon dioxide, aluminium oxide, iron oxide, and calcium oxide are the chemical compounds with the highest concentrations in their composition. Its chemical composition, on the other hand, changes depending on the type of coal and the furnace operation.

#### 2.1.2. Metakaolin (MK)

The metakaolin used in this research was produced by the calcination of commercially available kaolin clay [12] at low temperatures (heated up to 700 °C at a rate of 10 °C/min and maintained for 30 min.). As a result, the starting material was converted into an aluminosilicate source with strong pozzolanic activity. According to the XRF analysis, the metakaolin contains a high concentration of silicon and aluminium oxides (Table 1). The metakaolin used in the study had a particle size distribution of 9.2 μm (d50), a specific surface area of 16.8 m^2^/g and a bulk density of 0.22 ± 0.01 g/cm^3^.

#### 2.1.3. Mine Tailings (MT)

In this study mine tailings from barite mine were collected and subjected to calcination to remove water, organic matter and to improve reactivity. The parameters of the calcination process were the same as those used for the kaolin calcination. However, according to the oxide chemical composition analysis, the MT used in this study exhibit a high content of Fe oxides and a much lower content of SiO_2_ and Al_2_O_3_ than CA or MK. The mine tailings used in the study had a particle size distribution of 46 μm (d50), a specific surface area of 0.33 m^2^/g and a bulk density of 2.83 ± 0.01 g/cm^3^.

#### 2.1.4. Activator Solution

As an activator, an acid solution of o-phosphate (H_3_PO_4_) with 85 wt.% was diluted in distilled water to obtain a corresponding activation solution for an H_3_PO_4_/Al_2_O_3_ ratio of 1:1. According to previous studies [12,13], this ratio exhibits the best performance in terms of the compressive strength of obtained geopolymers.

Using the materials presented above, five types of geopolymers were synthetized. Table 2 shows the experimental mixes used in this study. The samples were designed to observe the influence of 50% by the mass addition of each raw material onto another. Accordingly, geopolymers with 100% of the solid component of coal ash (CA–geo) and metakaolin (MK–geo) were obtained, while because of the low geopolymerisation potential of mine tailings, a geopolymer with 100% mine tailing in the solid component could not be synthesized (the sample could not be cured at room temperature). In order to evaluate the addition of each component, three types of blended geopolymers have been designed, one from a mixture of metakaolin and coal ash (CAMK), one from a mixture of mine tailings and coal ash (MTCA) and another one with mine tailings and metakaolin (MTMK). Accordingly, these samples were obtained from a mixture of two of the raw materials at a 1 to 1 weight ratio.

All the mixtures have been activated with a phosphorous solution at a solid/liquid ratio of 0.9 (to assure workability). Moreover, to assure better homogeneity, the geopolymers with multiple aluminosilicate sources were mixed in a dry state for 2 min, and for 5 min after the liquid addition, according to the procedure presented in [18,19]. Accordingly, the mixtures were poured into 20 × 20 × 20 mm^3^ moulds, covered with a thin layer of plastic (to avoid moisture loss) and cured at room temperature (22 ± 2 °C) for 28 days before testing. The schematic representation of samples obtained is presented in Figure 2.

### 2.2. Methods

The thermal behaviour of the acid-activated geopolymers obtained in this study was evaluated through differential thermal analysis (DTA) combined with thermogravimetric analysis (TGA). Moreover, X-ray diffraction (XRD) was involved to analyse the phase transition following the geopolymerisation reaction.

#### 2.2.1. X-ray Diffraction

The mineralogical composition of the acid-activated geopolymers was analysed by XRD using X’Pert Pro MPD equipment (Malvern Panalytical Ltd., Eindhoven, The Netherlands). The CuKα radiation was recorded in the range of 5° and 60° 2θ using a single channel detector. The radiation was collected at a step size of 0.013°, while the copper X-ray tube was operated at a 40 mA current intensity and a 45 kV voltage.

The XRD analysis was performed on samples in a powder form, which were grounded after the curing period.

#### 2.2.2. Simultaneous Thermal Analysis

The transformations of phases and weight evolution was evaluated in the 25–600 °C temperature range using STA PT–1600 equipment (Linseis, Selb, Germany). The heating was performed in a static air atmosphere on samples lighter than 50 mg at a rate of 10 °C/min.

## 3. Results and Discussion

### 3.1. Phase Transition Analysis

The XRD patterns were evaluated using Highscore software v5.1, while the identification of the peaks was conducted using the PANalytical 2021 database. The peak search was realized considering only those with a minimum significance of 5.00, and a peak base with Gonio of 2.00, through the minimum second derivative method.

The XRD analysis of the CA samples (Figure 3) confirms the presence of multiple phases, such as quartz (96-900-9667), calcite (96-900-0967), anorthite (96-900-0362) and hematite-proto (96-900-2163). The detected hematite-proto contains Fe, H and O. Moreover, the XRD pattern of the metakaolin showed a high content of a typical amorphous structure. Compared to the CA pattern, the one specific to MK showed multiple peaks with high intensity. The most significant peak is positioned close to 25° (2θ) and corresponds to kaolinite (96-900-9231), the second peak, positioned close to 12.5° (2θ) corresponds to the same phase, while the peak close to 26.6° (2θ) corresponds to quartz (96-900-9667). Close to 35° (2θ) multiple peaks confirm the presence of muscovite (96-901-4938), while chloritoid (96-900-5444) was detected with the most significant peak close to 20° (2θ). The detected kaolinite contains Al, Si and O, the detected quartz contains Si and O, the detected muscovite contains K, Al, Si and O and the detected chloritoid contains Al, Fe, Mg, Si and O. As can be seen from Figure 3, in the case of MK, almost all of the identified peaks confirm the presence of kaolinite, quartz, muscovite and chloritoid.

In the case of MT, the XRD pattern confirmed the presence of multiple phases rich in Fe or Si. Accordingly, quartz (96-901-5023), hematite (96-901-5965), pyrite (96-900-0595), calcium cyclo-hexaaluminate (96-100-0040) and hydrazinium copper sulfide (96-430-7636) were detected.

Due to the reaction between the activator and the aluminosilicate source, the dissolution of aluminates and silicates occurred, resulting in disordered and amorphous silicophosphate (Si–P), aluminophosphate (Al–P) or silico-alumino-phosphate (Si–Al–P) gels. However, the differences between the XRD spectra of the raw materials and the spectra of the acid-activated geopolymers are low because the Si–P, Al–P and Si–Al–P phases are typically amorphous (Figure 4). The existence of quartz and kaolinite at the same position was evident in previous studies [24,25]. Moreover, it can be observed that, due to the geopolymerisation reaction, multiple peaks disappeared, such as the peak around 22° (2θ) and the one around 29.5° (2θ). In addition, the intensity of the peaks decreased significantly. However, one new peak can be observed around 40.5° (2θ).

The calcite disappearance from the CA after the reaction with the phosphoric acid, i.e., the peak around 28° (2θ), disappeared due to geopolymerisation. This could be attributed to brushite formation or to the following chemical reaction (Equation (1)):3Ca^2+^ + 2PO_4_^3−^ + H_2_O = 3CaO − P_2_O_5_ – H_2_O (C–P–H gel)(1)

In a phosphate acid environment, the Al oxides will be dissolute after the Ca compounds; therefore, the C–P–H gel will be formed before Al-P [10]. Therefore, by comparing the samples with MK against those without MK, it can be stated that the samples with higher Ca content will exhibit a lower setting time, due to the solubility differences between the divalent metals and the trivalent one.

The broad peak at 27° (2θ) indicates the formation of the berlinite phase (AlPO_4_) which has a similar XRD pattern to quartz and will confer high mechanical properties to the final geopolymer [10,26]. Moreover, by comparing the XRD pattern of the MK with that of MK–geo, it can be observed that all of the peaks between at 26° (2θ) and 35° (2θ) appeared due to the reaction between the acid and the raw material. Furthermore, the intensity of all the peaks specific to kaolinite have been reduced.

In addition, as observed in [27,28], multiple Al–O–P phases, such as phosphotridymite, phosphocristobalite, aluminium phosphate or aluminium phosphate hydroxide appear, but they overlap with the patterns of other phases. However, a hydrated form of aluminium phosphate confirmed as metavariscite was detected [29].

In the samples with mine tailings (Figure 5), the characteristic diffraction of anhydrite (CaSO_4_) disappeared, which indicates that the structures of CaSO_4_ were dissolute due to the phosphoric acid presence. Furthermore, the resulting Ca contributes to the formation of brushite or amorphous calcium phosphate [30] and as crystalline ettringite in Aft and Afm monosulfate [31,32]. In addition, the peak around 12° (2θ), which corresponds to hydrazinium copper sulfide, increased significantly in the MTCA geopolymer.

### 3.2. Thermal Behaviour Analysis

The TG-DTA analysis was used in this study to analyse the thermal behaviour of five types of geopolymers. Accordingly, it was observed that the addition of another aluminosilicate source influenced the metakaolin or coal ash-based geopolymers. By evaluating the heat flux and the mass evolution during the heating of the sample, the volatile compounds were eliminated, while the transition of different phases were observed. Moreover, by overlapping the TGA with the DTA curve, the mass loss or gain at specific temperatures could be correlated with the heat flux, to confirm the presence of a specific phase and its amount.

The samples analysed in this study showed multiple peaks on the DTA curves, which have been correlated with the water evaporation and oxidation processes. As can be seen in Figure 6, the samples without mine tailings exhibited similar behaviour, except for the samples CAMK which showed an endothermic peak around 220–260 °C. Moreover, the samples with mine tailings exhibited important oxidation reactions above 400 °C.

Moreover, as can be seen in Figure 6, the acid-activated geopolymers exhibit similar behaviour to the alkali-activated ones. Accordingly, the DTA curve of the coal ash-based geopolymer (CA–geo) shows multiple peaks, while the most significant one is the endothermic peak around 131 °C. This peak is correlated to the removal of the water molecules, which can exist as a free or chemical bond with the components from the geopolymers’ structure. Therefore, the use of phosphoric acid as an activator will lead to the development of a porous structure with zeolitic channels, which keep the water at temperatures much higher than evaporation. Firstly, the hygroscopic water is removed until 120 °C, this exists in the structure of the geopolymers due to their hygroscopicity. Secondly, up to 300 °C, the physically strong bond molecules of water are removed as follows: (i) up to 200 °C the crystallization water bounded in the structure during the crystal formation from the aqueous solution is created by mixing the aluminosilicate source with the activator; (ii) during heating between the 180 and 300 °C temperature range, the molecules from intracrystalline type or network type hydrogels are removed. In approximately the same temperature range, zeolitic water will be removed from the channels. The behaviour is similar also for MK-geo and CAMK, the peak temperature being changed in accordance with the amount of water and the size of the sample. Moreover, the metakaolin-based sample showed the largest peak as it has a higher amount of gel pores and zeolitic channels compared with the coal ash-based ones. Accordingly, the minimum value of the first peak was moved to higher temperatures, close to 167 °C. The blended (mix of two aluminosilicate sources, coal ash and metakaolin) geopolymer, CAMK sample, showed a broader peak into the water removal temperature range. This change could be correlated with the influence of the thermal behaviour on water removal at a low temperature from large pores specific to coal ash-based geopolymers overlapping next to water removal from small pores specific to metakaolin-based geopolymers.

By comparing the broadening of the first peak, it can be stated that the metakaolin-based geopolymers contain a higher quantity of water in small pores compared with those based on coal ash. These results are fully in line with the pore size distribution in geopolymers evaluated by NMR in previous studies, where the authors from [33] discovered a large amount of pores around 2.5 nm in metakaolin-based geopolymers, while in [34], the experiments on coal ash geopolymers showed that the first peak, on the relative pore size distribution, was positioned at higher relaxation time, i.e., the pores have a higher diameter.

As the samples are heated up above 300 °C, the chemically bound water will be removed. Accordingly, the fluctuations from the DTA curves correspond to the decomposition of acids, basics and neutral groups, which are formed between a metal and the OH groups. Considering the chemical composition of the raw materials, those can be Fe(III), Ti(IV), Si(IV), Na, K, P, Mg(II), Ca(II) and Al(III).

In the case of the blended geopolymer, the Na and Ca addition from the coal ash significantly affected the amount of water retained in Ca– or Al–silicate–hydrate channels and pores. This phenomenon can be due to the high concentration of Al brought into the system by the metakaolin. In other words, by mixing these two raw materials, the addition of these three elements (Na, Ca and Al) will have a significant impact on the condensation of C–A–S–H, C–S–H and N–A–S–H and, consequently, on the three-dimensional aluminosilicate network [35]. Accordingly, in the case of the CAMK sample, the water evaporation reaction from sodium–aluminosilicate hydrogel (N–A–S–H gel) is much more dominant in the hydrogels range (close to 185 °C), while the endothermic peak related to water removal from C–S–H and C–A–S–H structures can be observed around 240 °C. Moreover, in the case of acid-activated geopolymer, structures such as –Si–O–Al–O–P– gel will lose the water from the network in the same temperature range [10].

Above this temperature, no significant peak can be identified in the case of coal ash- or metakaolin-based geopolymers. However, when mine tailings are involved in the mixture, the thermal behaviour is significantly changed. Consequently, the separation of the endothermic peaks in the range of water evaporation is much clearer. Accordingly, the MTCA curve shows two minimum points, the first being close to 125 °C, and the second being around 155 °C. However, in the case of the metakaolin-mine tailings blended geopolymer, the DTA curve shows only one peak in this temperature range, which has the minimum point at 180 °C. Therefore, the MK presence contributed to a zeolite-like structure formation. Another significant difference between the samples with and without mine tailings is the appearance of the endothermic peaks above 400 °C.

By comparing the DSC curves of the studied samples with those of alkali-activated geopolymers presented in [36], it can be observed that in the 20–300 °C temperature range, the thermal behaviour is almost the same. Both types of materials exhibit water evaporation from large pores and zeolitic channels, followed by its removal from hydrogels.

The first exothermic peak with the maximum point around 460 °C for the MTMK sample, and around 480 °C for the MKCA sample, is the conversion of magnetite to hematite [37]. Considering the provenience of the mine tailings (dams), in the same temperature range, the exothermic reactions can be associated with humic acid disintegration [38]. Moreover, by heating sulfide ores such as pyrite in air atmospheres, the following chemical reaction can occur:4FeS_2_(s) + 11O_2_(g)→2Fe_2_O_3_(s) + 8SO_2_(g)(2)

The second exothermic peak appears due to the recrystallization of precipitated sulphate apatite at a lower temperature as a result of the transition of sulphate ions [39].

The weight loss during heating is related to the evaporation of water from the highly porous structure, which includes C–S–H and C–A–S–H formations that retain the activator in a liquid state even after multiple days of ageing. At higher temperatures, the weight loss is due to the decomposition of portlandite and other phases. Accordingly, the DTA curves exhibit endothermic peaks at corresponding temperatures, except for the samples with mine tailings addition which show high exothermic peaks in the range of 420–520 °C and 560–590 °C.

According to the TGA plots (Figure 7), up to 300 °C, the mass loss of the CA–geo sample was 12.7 wt.%, the mass loss of the CAMK sample was 16.2 wt.%, the mass loss of MK–geo was 18.7 wt.%, the mass loss of MTCA was 10.5 wt.% and the mass loss of MTMK was 14.4 wt.%. The mass change behaviour had the same trend as the first peak from the DTA curves, i.e., the samples with high water content showed a broader peak. Moreover, by correlating the type of raw material with the mass loss, it can be stated that the raw materials influence the water retention in the following order: metakaolin < coal ash < mine tailings. Therefore, this confirms that the samples with metakaolin have lower pores and channels which retain water at high temperatures.

In the 300–600 °C temperature range, the mass loss was very low for the CA–geo and MTCA samples which showed an extra decrease of 0.2 wt.%. However, in the same temperature range, the MTMK sample showed an extra mass loss of 2.1 wt.%, CAMK showed a loss of 2.3 wt.% and MK–geo showed a loss of 2.8 wt.%. Starting around 420 °C, with a maximum peak around 460 °C for the MTMK sample and 480 °C for MKCA, an exothermic reaction occurred. The weight gain corresponding to this oxidizing reaction was close to 0.5 wt.% for MTMK and 0 wt.% for MTCA. Accordingly, due to oxidation, only a small amount of oxygen remained in the sample, while other chemical elements evacuated.

## 4. Conclusions

This study examined the thermal behaviour and mineralogical composition of acid-activated geopolymers. Coal ash, metakaolin, mine tailings and different mixtures of these three aluminosilicate sources were used to synthesize silico-aluminophosphate geopolymers. Based on the experimental results reported, the following conclusions are drawn:In the 20–300 °C temperature range, the geopolymers obtained with H_3_PO_4_ acid exhibited similar thermal behaviour to those activated with a mix of NaOH and Na_2_SiO_3_,In the 400–600 °C temperature range, the geopolymers with mine tailings addition exhibited exothermic reactions, while those without mine tailings addition did not show significant phase transition,Up to 600 °C, the total mass loss of the Ca–geo was 12.9 wt.%, 21.5 wt.% for the MK and 18.5 wt.% for CAMK. The MT addition decreased the mass loss at 10.5 wt.% when mixed with CA and 14.4 wt.% when mixed with MK,The XRD analysis confirmed the formation of the ettringite phase in the geopolymers with MT addition and berlinite, brushite or metavariscite in those based on coal ash or metakaolin.

## Figures and Tables

**Figure 1 materials-15-00202-f001:**
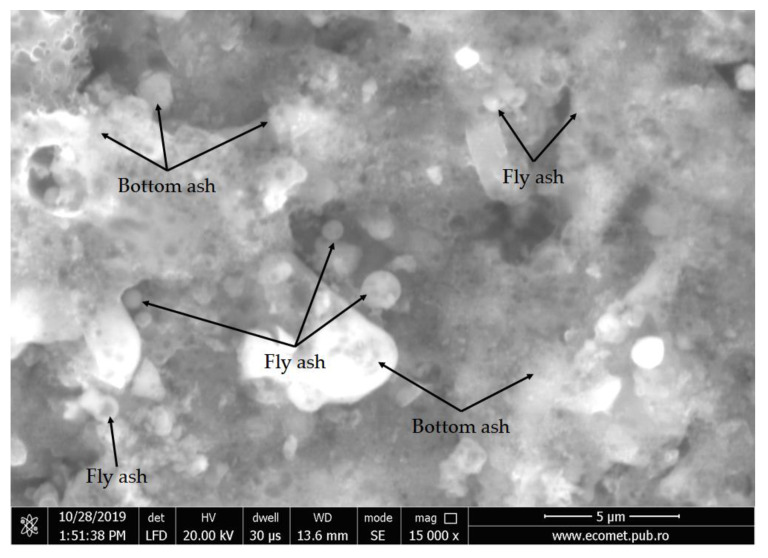
Coal ash morphology showing bottom and fly ash particles.

**Figure 2 materials-15-00202-f002:**
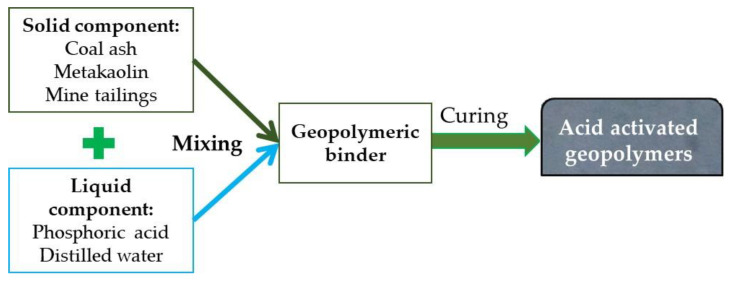
Process flow diagram of the obtained samples.

**Figure 3 materials-15-00202-f003:**
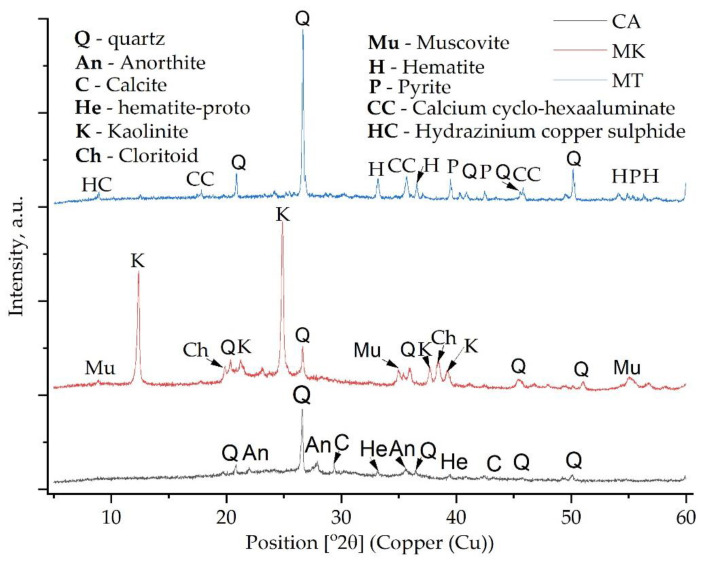
The XRD pattern of the raw materials used in this study.

**Figure 4 materials-15-00202-f004:**
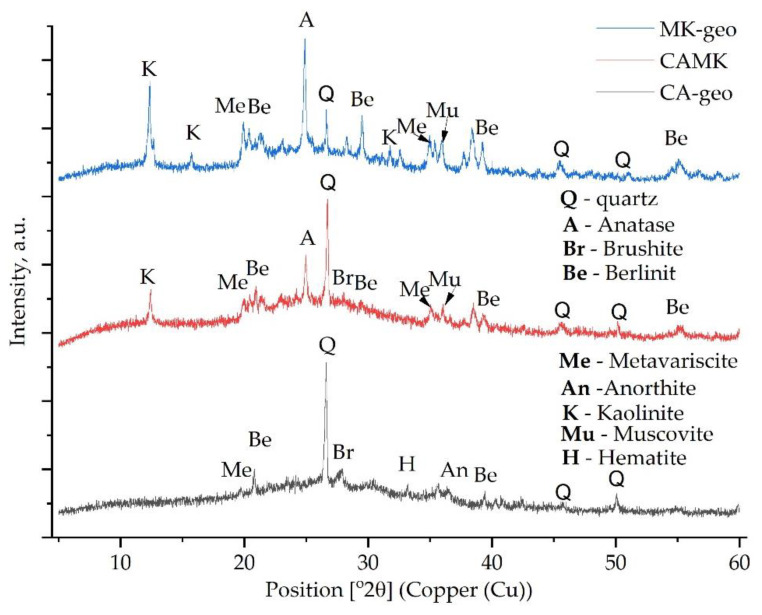
The XRD pattern of the acid-activated geopolymers without mine tailings.

**Figure 5 materials-15-00202-f005:**
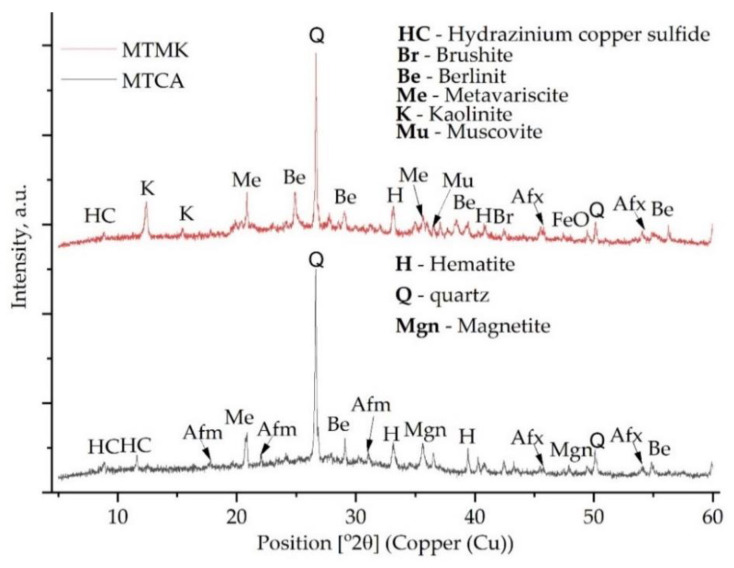
The XRD pattern of the acid-activated geopolymers with mine tailings.

**Figure 6 materials-15-00202-f006:**
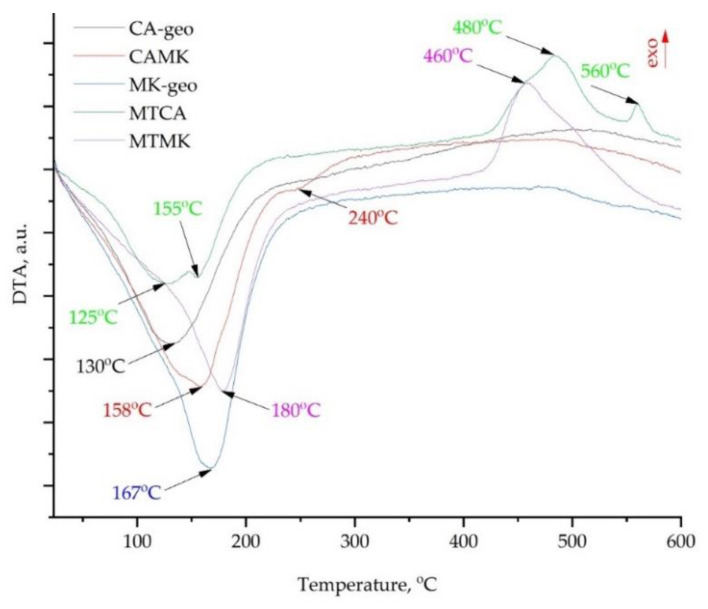
DTA plots of the studied geopolymers.

**Figure 7 materials-15-00202-f007:**
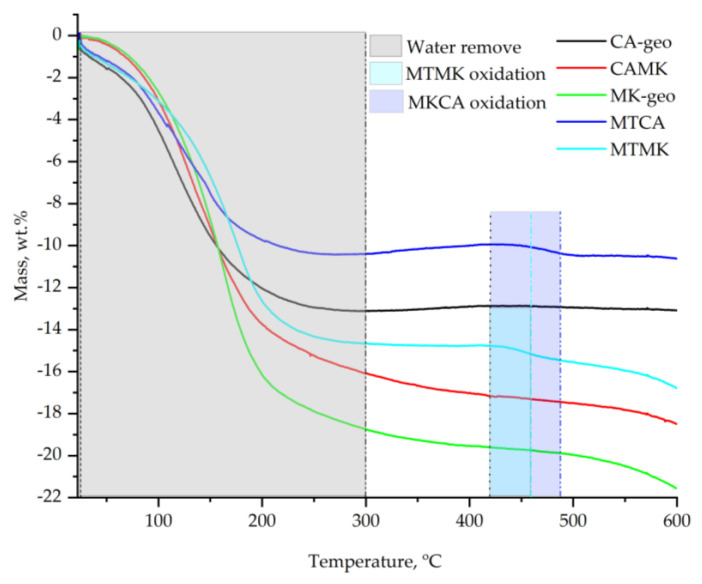
TGA plots of the studied geopolymers.

**Table 1 materials-15-00202-t001:** Oxide composition of raw materials, coal ash (CA), metakaolin (MK) and mine tailings (MT).

Sample	Oxide	SiO_2_	Al_2_O_3_	Fe_x_O_y_	CaO	K_2_O	MgO	TiO_2_	CuO	Na_2_O	P_2_O_5_	SO_3_	Oth.*	L.O.I.**
CA	%, wt.	46.1	27.6	9.8	6.2	2.3	1.9	1.3	0.0	0.6	0.4	-	0.3	3.5
MK	%, wt.	52.1	42.5	1.2	0.7	0.5	0.2	0.9	0.0	-	0.2	-	0.4	1.3
MT	%, wt.	16.2	2.6	38.9	0.4	0.6	-	0.2	0.5	-	0.3	11.4	0.9	28.1

Oth.*—oxides in a concentration lower than 0.1% (traces of S, Cl, Cr, Zr, Ni, Sr, Zn and Cu). L.O.I.**—Loss on ignition.

**Table 2 materials-15-00202-t002:** Mix design and parameters.

Sample Code	Coal Ash, wt.%	Metakaolin, wt.%	Mine Tailings, wt.%	Al/P Molar Ratio	Curing, °C
CA-geo	100	-	-	1	22 ± 2
MK-geo	-	100	-	1	22 ± 2
CAMK	50	50	-	1	22 ± 2
MTCA	50	-	50	1	22 ± 2
MTMK	-	50	50	1	22 ± 2

## Data Availability

The study did not report any data.

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
