# Peer review of "XRD and TG-DTA Study of New Phosphate-Based Geopolymers with Coal Ash or Metakaolin as Aluminosilicate Source and Mine Tailings Addition"

_materials, 2021, doi:10.3390/ma15010202_

Round 1
Reviewer 1 Report
This study aimed to produce acid-activated geopolymers on a laboratory level. The 5 mixtures contained metakaolin, fly ash and/or mine tailings. The idea of the work is of interest, as mostly the acid-activation of pure metakaolin is covered in the literature. However, the chosen methodology is not satisfying nor complete. Also, some of the key issues are not clearly explained.
- The main problem of this work is a scarce experimental section. I believe chemical, mineralogical and DTA/TG analyses are not enough to describe the obtained materials. The loss on ignition should be presented in the chemical analyses.
- The authors should bear in mind that the calcination of kaolin is done to produce highly amorphous material and that this is the reason for the enhanced reactivity of the precursors. It seems the authors did not understand the geopolymerization process. XRD results should include the quantity of the amorphous portion of the material. To obtain a geopolymer of satisfying properties, especially by using the acid-activation process, a great portion of the amorphous matter is needed. This might be the reason for a “low geopolymerisation potential of mine tailings”.
- XRD results should be presented before DTA/TGA, as thermal analyses are complementary to chemical and mineralogical content.
- It is not clear why only firing until 600 °C is done. The authors made 3 figures of DTA/TGA, while only one would be enough. The authors insisted on a hardly observing peak at 240 °C that is not important and might have been a noise. Besides, decarbonization is mentioned in the text, while the temperature needed for this reaction is not reached in the tests. Also, a peak at 130 °C or 140 °C (it is claimed differently at different places, the text and the figure) which is emphasised as a zone A1 is negligible. The first endothermic peak at 480 °C might be of the organic matter, which was not stated in the text (lines 307-308). Why DTG was not presented and discussed?
- It seems the samples were not shaped in a mould; it is not clear how the samples were prepared and left to cure for 28 days. Were they shaped, cured, and then milled before the analysis?
- No important mechanical property was observed, such as compressive strength, modulus of rupture, water absorption, bulk density. A few analytical techniques do not tell us about the quality of the product.
- The title should mention that also metakaolin is used in the mixtures.
- Some sentences are not clearly written; English must be improved.
- Many claims are not documented by a reference, nor compared to literature. They were just stated, such, for example: “During heating, the acid-activated geopolymers exhibited similar behaviour with those alkali-activated.” Which behavior is that, and where are the results?
- Some of the paragraphs contained plagiarism (lines 53-66, 115-130, 199-222, and 355-362).
- There are 2 tables named Table 1. The abbreviations of the mixtures are not the same throughout the Manuscript, should be shorter and clearer to the reader (1RM should be removed).
- Is the formation of zeolites during the geopolymerization desirable in these systems? Too long discussion on zeolites is contained in the DTA/TG analyses when, at the moment, it is not clear is their presence actually confirmed or not.
- This statement is not the truth: “According to the chemical composition of the raw materials, the weight percentage of silicon oxides is higher in MK than that in CA or MT, indicating quartz and free silica.”
Author Response
We would like to thank the reviewers for the meticulous and thorough review of this manuscript. We hope that our improvements will meet the requests of the reviewers and editors as closely as possible.
The point-by-point replies are attached.

Reviewer 2 Report
The manuscript "XRD and TG-DTA Study of New Phosphate-based Geopolymers with Coal-Ash as Aluminosilicate Source and Mine Tailings Addition" seeks to bring correlations and analogies related to microstructural analysis of geopolymer materials, a great research topic. Authors may publish this paper after minor corrections:
a) The use of characterization techniques such as "XRD and TG-DTA" is not recommended, I would suggest the use of analytical techniques or similar term;
b) The introduction is limited, there are few studies in the literature on the research topic, I suggest the use of recent research, which may help and show disadvantages even of geopolymeric materials, in addition to discussions of the results, only 38 references is not enough. I suggest reading and inserting the following works: 10.1016/j.conbuildmat.2021.122994; 10.1016/j.cscm.2021.e00802; 10.1016/j.cscm.2021.e00733.
c) The procedure for dosing geopolymer materials is not clear to readers, this should be better discussed and presented;
d) The texts present inside the images, as in Figs. 7, 8,9 and 10 are incomprehensible;
e) "According to the chemical composition of the raw materials, the weight percentage of silicon oxides is higher in MK than that in CA or MT, indicating quartz and free silica. Also, the K2O presence from CA, indicates illite mineral presence, and the high content of FexOy oxides from MT indicates Pyrite, Ferrite or Anhydrite, while MgO and CaO presence could be associated with dolomite presence" Explain this sentence further. Note that there are some claims that need further discussion with the international literature;
f) The conclusion needs changes and corrections, it must be objective and clear to the readers.
Author Response
Thank you very much for your time and effort in reviewing our manuscript. We hope the changes listed have made the manuscript suitable for publication and we look forward to your response. We have uploaded the paper with the highlighted changes.
The point-by-point replies are attached.

Reviewer 3 Report
1. The XRD anaylsis has major problems that must be improved. The coal fly ash and metakaolin has big amount of amorphous phases according to the Appendix A, when the author want to do the quantitative analysis, a Reference Intensity Ratio (RIR) method should be used for the material containing big amount of amorphous phases. Please design the experiment properly and revise XRD section.
2. There are asymmetric peaks in the DTA anaylsis, a deconvolution process is recommended to explain the differences in details between the materials.
3. Please provide the fineness or surface area of the coal ash, metakaolin and mine tailings.
Author Response
We are very grateful to the reviewer for his valuable comments on the manuscript. We have tried to address these as best we could, and the manuscript has been revised accordingly. Please, find attached our point-by-point answers.

Round 2
Reviewer 1 Report
The paper is now much improved and clear to the readers. It is ready to be published
Author Response
Thank you very much for your valuable recommendations and support. Your effort significantly improved our manuscript.
Reviewer 3 Report
The authors addressed most of my previous comments, but the there are a few points need to be improved.
- The amorphous phases in geopolymer is very important, the author should provide more discussion in details (i.e. compare the brand region and compare the indensity change, etc). FTIR is recommended to research the phase change as the author delete all the quantitative analysis in XRD.
- Please use weight loss in percentage in TGA analysis, and revise the manuscript accordingly.
Author Response
Thank you very much for your review of our manuscript, and we are very honoured to receive valuable opinions from Materials. The authors strongly agree with the opinions you presented. During this period, we have checked our manuscript again and made some modifications. According to the opinions, the authors improved the quality of the discussion related to the evaluation of the raw materials by XRD and TGA graph presentation. Thank you again for the valuable suggestions. Please find attached our point-by-point reply.
